# Positive Mental Health of Migrants in the UK during COVID-19: A Review

**DOI:** 10.3390/ijerph20227046

**Published:** 2023-11-10

**Authors:** Yasuhiro Kotera, Habib Adam, Ann Kirkman, Muhammad Aledeh, Michelle Brooks-Ucheaga, Olamide Todowede, Stefan Rennick-Egglestone, Jessica Eve Jackson

**Affiliations:** 1School of Health Sciences, University of Nottingham, Nottingham NG7 2TU, UK; yasuhiro.kotera@nottingham.ac.uk (Y.K.); olamide.todowede@nottingham.ac.uk (O.T.); stefan.egglestone@nottingham.ac.uk (S.R.-E.); 2Center for Infectious Disease Education and Research, Osaka University, Osaka 565-0871, Japan; 3Institut für Religionswissenschaft, University of Vienna, 1010 Vienna, Austria; habib.adambaba@gmail.com; 4College of Health, Psychology and Social Care, University of Derby, Derby DE22 1GB, UK; a.kirkman2@derby.ac.uk (A.K.); michelleclaireb@yahoo.co.uk (M.B.-U.); j.jackson2@derby.ac.uk (J.E.J.); 5Department of Psychiatry, Wiener Gesundheitsverbund, Klinik Donaustadt, Langobardenstraße 122, 1220 Vienna, Austria

**Keywords:** migrant health, positive mental health, COVID-19, literature review, UK migrants, under-researched areas, quality assessment, pandemic

## Abstract

COVID-19 impacted the mental health of many people in the UK. The negative impact was especially substantial among vulnerable population groups, including migrants. While research has focused on the negative aspects of mental health during the pandemic, the positive mental health of migrants in the UK during COVID-19 remained to be evaluated. This review aimed to identify literature that focused on positive mental health, and thematically synthesise the findings to understand what positive mental health approaches were employed to support specific outcomes during the pandemic for them to survive in this difficult time. Medline, Embase, and PsycINFO were searched using terms including “mental health”, “migrants”, and “COVID-19”. The Critical Appraisal Skills Programme checklist was used to assess the quality of the included studies. There were only two studies examining the positive mental health of UK migrants during this period. They describe approaches such as religious beliefs, passion for and acknowledgement of their job, learning new things, being physically active, social media, and social activities, producing outcomes such as inner peace, confidence, well-being, and a sense of belonging. The quality of the included studies was high. More research about positive mental health in migrants in the UK during the pandemic is needed.

## 1. Introduction

### 1.1. Mental Health Impact of COVID-19 in the UK

The 2019 coronavirus disease (COVID-19) has had a significant impact on the mental health of people around the world [1,2]. The individual stressors experienced during this time include the fear of contracting the virus itself, wanting to maintain the safety of loved ones, and a lack of social connection that could be experienced in usual activities [3,4,5]. In addition to these stressors, COVID-19 exacerbated health inequalities (i.e., “avoidable, unfair and systematic differences in health between different groups of people” [6]), resulting in higher mortality and morbidity rates among those in the most socially disadvantaged community groups [7]. There were also high rates of vaccine hesitancy (i.e., a delay in receiving or refusing vaccination despite it being available [8]) among Black ethnic minorities, driven by mistrust and safety concerns [9]. Specifically, in the United Kingdom (UK), inequalities in the social determinants of health widened during the pandemic [10]. The social determinants of health refer to factors that are not medical, but influential to health outcomes [11]. Those factors are often embedded in people’s daily lives, relating to how people are born, live, work, and age. Examples are economic policies, social norms, and political systems.

These challenges are thought to have contributed to increases in mental health challenges; the prevalence of anxiety increased by 26% and depression by 28% during the period of COVID-19-related government restrictions in the UK [12]. Even after the pandemic, mental health challenges continued. A systematic review and meta-analysis of longitudinal studies identified that clinically significant mental health symptoms remain high, despite the reduced risks of the virus and the lifted restrictions [13]. More severe mental health challenges were reported in the migrant population in the UK [14]. For example, the mental distress derived from reduced working hours due to the COVID-19 restrictions was higher among migrant working men than native-born working men. Understanding the mental health impact of the COVID-19 pandemic, specifically for vulnerable groups that are most at risk of the pandemic’s impact, is essential.

Certain protected characteristics have specifically been identified as relating to an increased risk of poorer mental health outcomes during the pandemic. These include being a woman, younger in age, and being part of an ethnically or sexually marginalised group [15]. Migrants in particular have been highlighted as having an increased risk of infection, but have been disproportionately represented among COVID-19 cases [16]. However, knowledge about the impact on the mental health of diverse migrant groups under pandemic conditions is underdeveloped and urgently required as a step to meet their needs.

### 1.2. Positive Mental Health

Positive mental health focuses on an individual’s functioning, quality of life, and well-being, aiming to achieve individual and community fulfilment [17]. Positive mental health is different from traditional mental health approaches that primarily focus on the prevention of and reduction in negative symptoms, such as depression, anxiety, delusion, and obsessions [18]. Positive mental health involves supporting positive emotions and functioning, which helps people manage their mental health in daily life, work productively, and have a fruitful social/community life [19]. Therefore, positive mental health can manifest a broader perspective (e.g., occupational and social domains of life) than positive psychology that focuses on the positive aspects of human experiences such as happiness and strengths [20]. Components of positive mental health are commonly thought to include the prevalence of positive emotions, high emotional and social intelligence, subjective well-being, and resilience [21]. Strengthening positive mental health can prevent negative mental health symptoms and facilitate recovery from these symptoms [22].

There are two major traditions within positive mental health: hedonism and eudaimonism [23]. The hedonic tradition focuses on positive emotions and moods, and the elimination of negative ones. The eudaimonic tradition focuses on living well and achieving one’s potential. In other words, hedonism can be characterised as short-term orientation. Hedonism relating positive mental health focuses on maximising pleasure and minimising pain for immediate gratification. On the other hand, eudaimonism can be characterised as long-term orientation. Eudaimonism relating positive mental health focuses on future wellness through personal growth, meaningful activities, and the clarification and realisation of your values. Because positive mental health includes both traditions, it is regarded as the presence of general emotional, psychological, and social well-being [24]. More recently, six attributes of positive mental health were identified: (a) problem solving and self-actualisation; (b) personal satisfaction; (c) autonomy; (d) interpersonal relationship skills; (e) self-control; and (f) prosocial attitude. The attributes of problem solving and self-actualisation refer to effectively controlling the environment around you, for example, successfully handling stressful situations, overcoming negative thoughts or good sleep hygiene. Further examples include demonstrating good problem-solving abilities and having continuous personal growth. The personal satisfaction attribute refers to the ability to have an optimistic outlook and be satisfied with our lives. The attribute of autonomy refers to personal security and having a sense of control over our lives. The interpersonal relationship skills attribute is the need to connect with others and the ability to empathise with others. Self-control refers to the ability to manage our own emotions, such as stress and conflict. The last attribute is prosocial attitude, and this refers to the connection to the community, and helping and supporting others [25]. This new aspect of mental health has been discussed and theorised to help people better understand what it is and what it is not.

Positive mental health remains an under-researched area; however, its importance was highlighted during the COVID-19 pandemic. Recognising the global mental health challenges during the pandemic, the World Health Organization emphasised the need for enhancing people’s mental health resources, instead of solely addressing mental health problems [26]. Unsurprisingly, numerous studies have examined negative mental health during COVID-19 [2,4,27,28]. In contrast, research about positive mental health is still underdeveloped.

### 1.3. Migrant Mental Health

Migrants in the UK, in particular refugees and asylum seekers, experience mental health challenges. The causes for their mental health challenges include the process of moving, assimilation into their new country, enculturation into their new communities, and the experience of loss, grief or sadness for the loved ones that they have left behind in their home countries [29]. Such experiences during non-emergency times are already challenging, and with the presence of a global pandemic such as COVID-19, such challenges can be exacerbated. Because the COVID-19 pandemic impacted individuals in different ways, and arguably each person’s experience is unique to them, it is almost impossible to suggest that a universal experience of COVID-19 exists [30].

Migrants who move to the UK are not a monolithic group, coming from a variety of countries and representing a variety of cultures and races: they have had different experiences. In 2022, 1.2 million people migrated to the UK, whilst 557,000 individuals emigrated from the UK [31]. Net migration (the number of migrants minus the number of emigrants) in the UK has an increasing trend [31]. The UK was ranked as the second largest estimated net migration country in Europe in 2023 (*n* = 165,790) following Ukraine (*n* = 1,784,718) [32]. A large-scale (*n* = 1698) migrant interview study in a South East London community identified that discrimination based on migrant status was prevalent, negatively impacting migrant mental health [33]. Other UK migrant studies reported that cultural stigma (e.g., disclosing private matters to others can violate one’s family honour [34]) and language barriers hinder mental health help-seeking among migrants from Central and Eastern Europe [35]. Nepali and Iranian migrants in the UK reported mental distress derived from a lack of cultural understanding (e.g., what kind of emotional expression and non-expression are considered normal in one culture [36]) by healthcare workers [37]. Pregnant migrant women also suffer from poor mental health, which is attributable to stigma and communication difficulties (e.g., a cultural norm about what can be directly expressed or what needs to be discussed rather than privately [38]) [39]. Diverse negative mental health impacts experienced by diverse migrants in the UK have been reported. The positive mental health of migrants in the UK has been less reported. This is concerning because the mental well-being among people in the UK was poorer than in many other countries [40,41].

### 1.4. Study Aim

This literature review aimed to identify empirical study articles about the positive mental health of migrants in the UK during the COVID-19 pandemic, and synthesise the evidence reported in these articles. We investigated the quantity and quality of studies that focused on migrants’ positive mental health during the pandemic in the UK, and identified what positive mental health outcomes were evaluated along with the relevant positive mental health approaches used. Because we were aware that it was an under-researched area, we approached this review focusing on migrants in the UK in general, while noting above that migrants are not a monolith.

## 2. Methods

The design of this review is a general literature review. Our research questions (RQs) were “RQ1: How many studies have been conducted to evaluate positive mental health of migrants in the UK during the COVID-19?”, “RQ2: What positive mental health outcomes and approaches were reported in these studies?”, and “RQ3: What is the quality of these studies?”.

The inclusion criteria for articles were clarified using the PEO (population, exposure, and outcomes) framework: (P) migrants in the UK, (E) COVID-19, and (O) positive mental health, including the quality of life, well-being, fulfilment, and resilience. These outcomes were identified from systematic reviews on positive mental health [42,43]. Additionally, empirical papers that were quantitative, qualitative or both (article type), in English (language), and published between December 2019 and May 2023 (published time) were included. Table 1 presents our research questions and inclusion criteria.

Medline, Embase, and PsycINFO were searched because these databases include mental health and psychology research. The search terms included “migrant”, “UK”, “COVID”, and positive mental health outcomes, using Boolean operators (e.g., migrant AND UK AND COVID AND resilience). The included papers were reviewed by three co-authors (HA, AK, MA, and OT) who were specialists in mental health and migrants. The findings were thematically synthesised by YK [44]. The synthesised findings were reviewed by all co-authors for coherency and clarity.

The quality of the included articles was assessed using the CASP (Critical Appraisal Skills Programme) checklist [45] by identifying the risk of biases. The scoring was 0–4 for high risk, 5–8 for medium risk, and 9–12 for low risk [46]. These assessments were performed independently by two co-authors (YK and AK), and a meeting was to take place if there was a disagreement between the two.

## 3. Results

### 3.1. Included Articles (Responding to RQ1)

Two articles were found eligible [47,48]. The two articles were qualitative studies, focused on migrants from Pakistan, China, Iran, and Italy. Saleem et al.’s study [47] investigated 10 physicians from Pakistan (six males and four females). Their age was not reported. Yen et al.’s study [48] investigated 60 migrants, comprising those from China, Iran, and Italy, 20 each. Table 2 summarises the two articles.

### 3.2. Positive Mental Health Outcomes and Approaches (Responding to RQ2)

Saleem et al.’s study [47] investigated 10 physicians from Pakistan. During COVID-19, they experienced an increase in inner peace and confidence, which was supported by their (a) spiritual beliefs and (b) passion for and acknowledgement of their job.

Yen et al.’s study [48] investigated 60 migrants from China, Iran and Italy with different occupations. They reported that (a) learning new things, (b) being physically active, (c) social media, and (d) social activities were helpful in supporting their well-being and sense of belonging.

In sum, four positive mental health outcomes were reported by migrants in the UK during the pandemic. These were inner peace, confidence, well-being, and a sense of belonging. Six approaches were reported to support these positive mental health outcomes: (a) religious beliefs, (b) passion for and acknowledgement of their job, (c) learning new things, (d) being physically active, (e) social media, and (f) social activities.

### 3.3. Study Quality (Responding to RQ3)

The risk of biases for the two studies was low (9/12 [47] and 10/12 [48]), suggesting high quality. “Data collection addressed research issues” was met in Yen et al.’s study [48], but was not certain (“CT” = Cannot Tell) in Saleem et al.’s study [47].

The initial assessments by the two reviewers were similar apart from one item in Yen et al.’s study, and one item in both studies. “Clear statement of aims” in Yen et al.’s study [48] was initially marked “Y (Yes)” and “N (No)”, but after discussion, both reviewers identified a clear statement, thus agreed as “Y”.

In both studies, the “Researcher–participant relationship considered” was initially marked “Y” and “N”, but after discussion, the two reviewers agreed on “N”. In both studies, an author network with snowballing referrals was used to recruit participants. We judged this was appropriate as migrant research often struggles to attain many participants [49]. However, the weaknesses of these recruiting methods included participants’ homogeneity, as participants are from the author’s network and extended network [50] (e.g., participants in one religious group may feel easier to introduce others in the same religious group to the researchers; participants in some cultures feel easier to introduce female friends than male friends due to mental health shame in males [51]). Moreover, after an interview, the participants gained knowledge about the interviewers. This also might have impacted the choice of which friends to introduce to them, e.g., if a participant perceived that the interviewer supported one political party, they might not have chosen their friend who supported the opposing party. Both studies did not report sufficiently on how they attempted to address these possible problems. The lack of researcher–participant relationship consideration presents this weakness. Rigorous analysis was not reported in neither study, for example, whether the researchers critically examined their own role, potential bias and influence during the analysis, and the selection of data for presentation. Table 3 presents the risk of bias assessment.

Taken together, our research questions were answered. For RQ1, the number of studies that focused on the positive mental health of migrants in the UK during COVID-19 was small, with only two studies. For RQ2, four positive mental health outcomes and six approaches were reported. These outcomes were inner peace, confidence, well-being, and a sense of belonging. The approaches were (a) religious beliefs, (b) passion for and acknowledgement of their job, (c) learning new things, (d) being physically active, (e) social media, and (f) social activities. For RQ3, the quality of the studies was high. Table 4 presents a summary of our findings.

## 4. Discussion

This review identified studies that focused on positive mental health during COVID-19, the outcomes and approaches, and assessed the quality of the studies. Two studies were included, targeting the positive mental health outcomes of inner peace, confidence, well-being, and a sense of belonging. These outcomes were supported by religious beliefs, passion for and acknowledgement of their job, learning new things, being physically active, social media, and social activities. The quality of the included studies was high.

First, we were able to find only two articles that were eligible according to our criteria. Considering the current emphasis on negative mental health, this was not surprising. Moreover, due to the negativity biases, during the emergency time, people might have been more susceptible to negative mental health than positive mental health. In a study on exploring self-compassion (positive mental health) and self-criticism (negative mental health), participants responded more strongly to self-criticism than self-compassion [52]. During the pandemic, people’s sensitivity towards negative mental health was heightened. Media, including social media, aggressively reported the negative impacts of a restricted life during the pandemic [53]. Stirring up people’s anxiety by media was considered purposeful in some contexts, as more access would benefit media companies [54]. Education about how to use media was highlighted in some countries during and/or after the peak of COVID-19 [55]. Universities re-introduced the use of media and social media to students as part of student well-being strategies [56]. These factors may indicate why the research about negative mental health is still mainstream compared with positive mental health. As highlighted in recent research [57], our findings suggest that more investigation into positive mental health is needed.

Second, the positive mental health outcomes of inner peace, confidence, well-being, and a sense of belonging, and the positive mental health approaches of religious beliefs, passion for and acknowledgement of their job, learning new things, being physically active, social media, and social activities were identified. These results are similar to reports on migrants’ mental health across different countries in Europe. A study in Norway indicated that the availability of social support for migrant workers reduced mental health issues among their populations [58]. Among Middle Eastern male Muslim immigrants in Germany, having a stronger religious belief was strongly associated with better mental health [59]. However, our review did not evaluate different groups in the UK migrant population. For example, a study in Canada showed that some positive mental health outcomes were reduced in refugees compared to other migrants [60]. The German study noted above [59] also reported that the changes in positive mental health outcomes were different between sub-groups of the studied migrants. These sub-group investigations will be helpful for UK migrants, as they can inform an alternative approach for reducing mental health problems such as depression, anxiety, loneliness, and post-traumatic stress disorder that are common among migrants [61]. Moreover, such targeted support for migrants may assist in reducing mental health inequalities among this population [62].

The similarities in the four positive mental health outcomes identified in this review may be that they were related to the soothing of our mind, which is one of the three emotional regulatory systems [63]. The soothing of the mind is about safeness and contentment, and is regarded as the most important for our mental well-being compared with the other two, i.e., the threat mind and the drive mind. During the uncertain time of COVID-19, migrants in the UK might have accessed these soothing-related positive mental health constructs and acknowledged the usefulness of them. The positive mental health approaches identified in this review were somewhat fundamental approaches to good mental health. Many people already know the positive mental health impacts of these approaches. This suggests that migrants might have been more appreciative of changes in their mental health during COVID-19; therefore, they felt those approaches were especially helpful to their daily life. The usefulness of an inner shift or reframing has been reported in COVID-19 studies [64,65]. Our findings suggest that migrants in the UK also had similar experiences.

Third, although there were only two studies, the quality of the included studies was high. Despite the challenges of the pandemic, the included studies addressed many of the assessment items (Table 2). One possible reason for this may be that in order to recruit the migrant population, who are often marginalised in society, the researchers might have approached the migrant population very carefully. Methodological work about migrant research recommends that multiple recruitment strategies be used [49]. These research practice guidelines might have contributed to the quality of the assessment.

In both studies [47,48], “Researcher–participant relationship considered” and “Rigorous data analysis” were unaddressed issues in the quality assessment. A challenge with the “Researcher–participant relationship considered” was, as discussed above, associated with their recruitment methods. For “Rigorous data analysis”, critical reflection of their own role, and how that might have influenced the research process and findings were not sufficiently reported. For example, reporting the reflexivity of the researchers could have been one way to address this issue [66]. Future research about migrants’ positive mental health in the UK needs to address these weaknesses.

While this review offers helpful insights, its limitations need to be noted. First, the amount of eligible studies was small. While the small quantity of eligible studies may indicate a lack of attention to positive mental health during COVID-19, more studies would have helped to yield more generalisable insights. Relatedly, the language was limited to English. This makes sense in a way, as our focus was on migrants in the UK. However, positive mental health studies during the pandemic, which affected people around the world, in other languages may have relevance to the UK. These studies might have been missed. Second, the co-authors of this paper are mental health researchers, practitioners, educators, and/or service users. Perspectives from people who are not in the mental health field were missed. Third, the definition of positive mental health is still being debated. Many different definitional emphases have been introduced such as maturity, positive emotions, and socio-emotional intelligence [67,68]. Although our findings can help inform the definition, they do not have strong definitional implications. Lastly, the mental health impact of the pandemic is still considered ongoing. New positive mental health studies associated with COVID-19 may be published. These are out of the scope of this review. Positive mental health researchers need to be aware of new COVID-19 positive mental health studies.

## 5. Conclusions

This review focused on the positive mental health of migrants in the UK during the COVID-19 pandemic. Our review identified that (1) there were only two studies focused on the positive mental health of migrants in the UK during COVID-19; (2) migrants in the UK used positive mental health approaches, such as religious beliefs and social activities, in order to improve positive mental health outcomes, such as inner peace and confidence; and (3) the quality of these studies was high. Future reviews could include more positive mental health COVID-19 studies (e.g., by including other languages) to further inform a wider perspective of positive mental health during the pandemic.

## Figures and Tables

**Table 1 ijerph-20-07046-t001:** Research questions and inclusion criteria for this review.

Research Questions (RQs)	RQ1: How Many Studies Have Been Conducted to Evaluate Positive Mental Health of Migrants in the UK during the COVID-19? RQ2: What Positive Mental Health Outcomes and Approaches Were Reported in These Studies? RQ3: What Is the Quality of These Studies?
Population (P)	Migrants in the UK
Exposure (E)	COVID-19
Outcomes (O)	Positive mental health including quality of life, well-being, fulfilment, and resilience
Other criteria	
Article type	Empirical papers that were quantitative, qualitative or both (reviews, commentaries, and theoretical papers were excluded)
Language	English
Published time	December 2019 to May 2023

**Table 2 ijerph-20-07046-t002:** Characteristics of the two included studies.

Authors	Published Year	Study Design	Sample (Sex, Age)	Outcome	Approach
Saleem et al. [47]	2021	Qualitative	10 Pakistani physicians (6 males and 4 females)	Inner peace	Religious beliefs
				Confidence	Passion for and acknowledgement of their job
Yen et al. [48]	2021	Qualitative	60 migrants (29 males and 31 females: Age 39 (21–67) years): China (10 males and 10 females: 44 (26–58)), Italy (10 males and 10 males: 37 (29–55)), and Iran (9 males and 11 females: 36 (21–67)).	Well-being	Learning new things
				Well-being	Being physically active
				Sense of belonging	Social media
				Sense of belonging	Social activities

**Table 3 ijerph-20-07046-t003:** Risk of bias assessment.

Qualitative Studies	Clear Statement of Aims	Appropriate Methodology	Appropriate Research Design	Appropriate Recruitment	Data Collection Addressed Research Issues	Researcher–Participant Relationship Considered	Ethical Issues Considered	Rigorous Data Analysis	Clear Statement of Findings	How Valuable is the Research? (0–3)	Score (0–12)
Saleem et al., 2021 [47]	Y	Y	Y	Y	CT	N	Y	N	Y	3	9
Yen et al., 2021 [48]	Y	Y	Y	Y	Y	N	Y	N	Y	3	10

Y = Yes, N = No, CT = Cannot Tell, NA = Not Applicable.

**Table 4 ijerph-20-07046-t004:** Summary of our findings.

Research Question	Findings
RQ1: How many studies have been conducted to evaluate positive mental health of migrants in the UK during the COVID-19?	Two studies.
	[47]
	[48]
RQ2: What positive mental health outcomes and approaches were reported in these studies?	Outcomes:
	inner peace, confidence, well-being, and a sense of belonging.
	Approaches:
	religious beliefs, passion and acknowledgement regarding their job, learning new things, being physically active, social media, and social activities
RQ3: What is the quality of these studies?	High: 9/12 and 10/12 (low risk of biases) using the CASP checklist

## Data Availability

Data are contained within the article.

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
