# Peer review of "Positive Mental Health of Migrants in the UK during COVID-19: A Review"

_ijerph, 2023, doi:10.3390/ijerph20227046_

Round 1

Reviewer 1 Report

Comments and Suggestions for Authors

The review presented responds to a very interesting and relevant topic for the scientific community and society. The development of the work adjusts to an appropriate methodology.

However, important limitations are detected:

- It is identified as a review, not a systematic review or meta-analysis, due to the number of publications obtained, limiting the scope and interest of the manuscript.

- The motivations for focusing exclusively on the work in the UK are not adequately justified. We miss a first objective with a global study, and then focusing on this country.

- In the introduction section reference is made to positive mental health, but the differences with respect to positive psychology are not adequately justified.

- Some terms related to positive mental health are identified (quality of life, well-being...) as well as the results (inner peace, sense of belonging). However, no type of initiatives are presented that serve as guidance for future review work including, for example, other descriptors. All this with the aim of being published in a prestigious journal.

Author Response

Reviewer 1’s comment 1

The review presented responds to a very interesting and relevant topic for the scientific community and society. The development of the work adjusts to an appropriate methodology.

However, important limitations are detected:

- It is identified as a review, not a systematic review or meta-analysis, due to the number of publications obtained, limiting the scope and interest of the manuscript.

Authors’ response 1-1

Thank you for your thoughtful and helpful comments. We agree with your point. While that by itself is an important finding that research on this topic is under-developed, it is indeed a limitation of this review. It is acknowledged and noted in the limitation section.

Reviewer 1’s comment 2

- The motivations for focusing exclusively on the work in the UK are not adequately justified. We miss a first objective with a global study, and then focusing on this country.

Authors’ response 1-2

In line with your comment, now the justification for why we focus on the UK is strengthened: many migrants and the low mental wellbeing in the UK relative to other countries.  Moreover, a weakness of this exclusive focus is noted in the limitation.

Reviewer 1’s comment 3

- In the introduction section reference is made to positive mental health, but the differences with respect to positive psychology are not adequately justified.

Authors’ response 1-3

In line with your comment, now the difference between positive psychology and PMH is clarified, informing the broader perspective of PMH.

Reviewer 1’s comment 4

- Some terms related to positive mental health are identified (quality of life, well-being...) as well as the results (inner peace, sense of belonging). However, no type of initiatives are presented that serve as guidance for future review work including, for example, other descriptors. All this with the aim of being published in a prestigious journal.

Authors’ response 1-4

Thank you for highlighting an important point. Indeed, a clear definition of PMH that distinguishes it from other descriptors has not been established. Our findings can inform the definition, but did not have strong implications. This is now added to the limitations.

Reviewer 2 Report

Comments and Suggestions for Authors

Dear Authors,

Thank you for the opportunity to review your manuscript. I have no objections except that in the method chapter you should specify a literature search strategy using Boolean operators to enable reproducibility of the research in the future.

Author Response

Reviewer 2’s comment 1

Dear Authors,

Thank you for the opportunity to review your manuscript. I have no objections except that in the method chapter you should specify a literature search strategy using Boolean operators to enable reproducibility of the research in the future.

Authors’ response 2-1

Thank you for your encouraging comment. Now Boolean operators are added to Methods.

Reviewer 3 Report

Comments and Suggestions for Authors

I have pointed out some minor addition of references to make the article much more a good read. There are no specific comments, however, I wanted the authors to set up a context using those references so that readers get to know the background about Covid-19 and how it impacted all the minors or the vulnerable population across the globe. Also, how the vaccines were dealt with in the developing and developed countries when the authors spoke about vaccine hesitancy. 

Comments on the Quality of English Language

English is ok. Some minor editing would enhance the quality of the paper. 

Author Response

Reviewer 3’s comment 1

I have pointed out some minor addition of references to make the article much more a good read. There are no specific comments, however, I wanted the authors to set up a context using those references so that readers get to know the background about Covid-19 and how it impacted all the minors or the vulnerable population across the globe. Also, how the vaccines were dealt with in the developing and developed countries when the authors spoke about vaccine hesitancy.

Authors’ response 3-1

Thank you for your feedback, capturing the strengths of our paper. Additional clarification and references are added in Introduction.

Round 2

Reviewer 1 Report

Comments and Suggestions for Authors

The manuscript has been modified following the reviewer's recommendations. Therefore, unless there is better judgment on the part of the Journal, the manuscript is accepted under the current terms.